# Low Maternal Care and Protection and Body Image Dissatisfaction as Psychopathological Predictors of Binge Eating Disorder in Transitional-Age Youth

**DOI:** 10.3390/nu17172737

**Published:** 2025-08-23

**Authors:** Emanuela Bianciardi, Rossella Mattea Quinto, Ester Longo, Valentina Santelli, Lorenzo Contini, Alberto Siracusano, Cinzia Niolu, Giorgio Di Lorenzo

**Affiliations:** 1Department of Systems Medicine, University of Rome Tor Vergata, 00133 Rome, Italysantelli.valentina@gmail.com (V.S.);; 2Department of Human Sciences, European University of Rome, 00163 Rome, Italy; 3Department of Mental Health, Asl Roma 6, 00041 Rome, Italy; 4IRCCS Fondazione Santa Lucia, 00179 Rome, Italy

**Keywords:** binge eating disorder, body image, emerging adulthood, maternal care, attachment styles, alexithymia, transitional age

## Abstract

**Background**: Binge eating disorder (BED) frequently arises during the transitional age (18–25 years), a critical developmental period characterized by challenges in autonomy, identity formation, and interpersonal functioning. This study investigated psychopathological predictors of BED risk in this age group, with particular focus on parental bonding, attachment style, body dissatisfaction, alexithymia, and depressive symptoms. **Methods**: A total of 287 participants aged 18–25 years completed the Binge Eating Scale (BES), Beck Depression Inventory-II (BDI-II), Body Shape Questionnaire (BSQ), Toronto Alexithymia Scale (TAS-20), Attachment Style Questionnaire (ASQ), and Parental Bonding Instrument (PBI). Sociodemographic information and body mass index (BMI) were also collected. **Results:** Compared with non-BED risk groups, individuals at risk of BED exhibited significantly higher BMI, greater alexithymia, higher body dissatisfaction, more insecure attachment patterns, and lower recalled paternal and maternal care. Hierarchical binary logistic regression revealed that the final model explained 56.1% of the variance (Nagelkerke R^2^) and correctly classified 92.1% of cases. Significant predictors of BED included body dissatisfaction, elevated BMI, low maternal care, and low maternal protection. **Conclusions**: This study is the first to examine BED risk factors specifically during the transitional age. Findings indicate that body image dissatisfaction, higher BMI, and inadequate maternal emotional care and protection are salient predictors at this life stage. Preventive interventions should integrate parental psychoeducation, nutritional guidance, and therapeutic strategies addressing both eating disorder symptoms and attachment-related difficulties to reduce BED onset and improve psychosocial outcomes in emerging adults.

## 1. Introduction

Binge eating disorder (BED) is a severe and disabling disorder causing mental distress, loneliness, and medical problems. It was previously reported to develop around the age of 40 [1], but recently, an earlier onset and an alarming increase in its prevalence and severity have been reported [2]. BED mostly begins in the transitional age from adolescence to adulthood, between the ages of 18 and 25, with a female to male ratio of 1:2, 1:6 [3]. In the under 30s, the lifetime prevalence of BED ranges from 0.6 to 6.1% in females and from 0.3 to 0.7% in males [4]. Studies conducted in Italy have reported that the lifetime prevalence of BED is 0.7% [5]. This earlier onset is noteworthy, as a large-scale meta-analysis of epidemiological studies conducted up to 2020 indicates that approximately 75% of mental disorders begin before the age of 25 [6] and the emerging adulthood is a major life transition probably representing the most unstable period of their lifespan [7]. On the one hand, the young adult begins to rely on himself, since he is not subject to parental authority like a child, on the other hand, he does not yet have the family and work stability of an adult, causing him to oscillate between worries and uncertainties. [8]. Hence, at this age, BED may be complicated by psychiatric comorbidities, but early intervention can prevent a more chronic disease and the risk of serious physical and psychopathological consequences [9,10]. In emerging adults, the literature on BED has identified body mass index, depressive symptoms, low self-esteem, and body dissatisfaction as risk factors [11,12], while other studies have highlighted the relevance of emotion dysregulation and impulsivity [13].

The “interpersonal vulnerability model of binge eating” [14] describes binge eating as a maladaptive behavior learned as a defensive measure to cope with negative effect caused by interpersonal dynamics, such as in adulthood, peer and romantic relationships and, in childhood, bonding with unsupportive and conflictual parents [15]. Therefore, the feeling in between autonomy and dependence on parents is unique at this stage of life and the type of parental bond may play an important role in the young adult’s psychopathological risk [16]. Interpersonal relationships are also a priority and a challenge at this age; changes in romantic relationships are the rule and for vulnerable emerging adults, peer comparisons and social expectations contribute to low self-esteem, and dissatisfaction with body image, all central factors in feeding and eating disorders (FEDs) [17]. Attachment theory was introduced by John Bowlby as a psychobiological system that describes how the quality of early parent–child interactions shape expectations, emotions, and behaviors in adult close relationships [18]. Insecure adult attachment style confers vulnerability to various diseases [19,20], including eating disorders, through the interplay of biological and psychopathological factors [21]. Nonetheless, to our knowledge, its association with BED has not been specifically investigated in the transitional age, and only a few authors have examined the role of parental bonding.

Parental bonding has been defined as the behavioral and emotional style towards the child [22] and can be assessed using the Parental Bonding Instrument (PBI), which separately describes the person’s recall of the maternal and paternal bond in the first 16 years of life outlining two dimensions: care and protection [23]. Care refers to the perception of emotional warmth and empathy. Protection describes closeness and encouragement of appropriate independence and autonomy [24].

Warm and empathic parenting has been shown to promote psychological development, whereas problematic parenting contributes to individual’ mental disorders [25,26]. To date, few studies have examined the association between parenting styles and FEDs symptoms [27], with most focusing on anorexia nervosa and bulimia nervosa, and recruiting samples of adolescents or adults separately. Specifically, low-caring, and overprotective parenting have been associated with anorexia, bulimia [28], and binge eating in pubertal girls [29]. This suboptimal parenting contributes to BED by undermining autonomy and self-esteem, thereby affecting food choices and the ability to resist binge eating impulses [30], with these vulnerabilities further intensified by heightened impulsivity and increased parental expectations that are typical of adolescence [31]. However, the influence of parental bonds on young adults—who are no longer as dependent on their families as adolescents, yet not fully autonomous as adults—has not been investigated.

We hypothesize that, in individuals aged 18–25 years, parental bonding influence the likelihood of having binge eating disorder in ways that differ from adolescence, and that this association remains significant even after controlling for relevant aspects in young adults such as dissatisfaction with body shape, difficulties in recognizing and communicating emotions, depression, and attachment style in close relationship.

## 2. Materials and Methods

### 2.1. Participants

A total of 287 young adults (72 males and 215 females; M_age_ = 23.21, SD = 1.6), aged between 18 and 25 years, who responded to an ad hoc online survey, were included in this cross-sectional study.

The 18–25 age range was selected to capture the transitional phase of emerging adulthood, a developmental stage characterized by ongoing emotional, social, and relational connections with the family of origin despite legal adulthood.

Data were collected between September and December 2024 through an anonymous, self-administered online survey developed using Google Forms^®^ (Google LLC, Mountain View, CA, USA). Participants were recruited using a convenience and snowball sampling strategy. The survey link was disseminated via social networks of university students and promoted through posters displayed in supermarkets in Rome, Italy, to reach a broad sample of emerging adults in the general population.

Online consent was obtained from the participants prior to data collection; they were allowed to terminate the survey at any time. The local Institutional Review Board approved all study procedures. The study was carried out in accordance with the 1964 Helsinki Declaration.

Inclusion Criteria were age between 18 and 25 years, provision of informed consent prior to survey participation, residence, study, or employment in Italy, ability to understand Italian and complete the questionnaire independently, completion of all required measures, and sociodemographic form, single submission per participant, verified through quality control procedures. Exclusion Criteria were age below 18 or above 25 years, absence of informed consent, incomplete survey responses or missing data on key variables, duplicate submissions or repeated entries identified through quality checks.

### 2.2. Measures

All participants completed the ad hoc survey comprising socio-demographic variables and psychometric questionnaires. Based on the literature on BED, we collected the following sociodemographic data: gender, age, educational level, marital status, and regular use of prescribed medication as an indicator of medical comorbidity [32]. Self-reported height and weight were used to calculate the BMI.

The Binge Eating Scale is a widely used self-report questionnaire designed to assess the severity of binge eating behaviors and identify individuals at risk for BED. It measures both behavioral and emotional symptoms associated with binge eating episodes. Each item presents four statements ranked in increasing severity (scores range from 0 to 3). A cut-off of 17 is used to identify moderate-to-severe binge eating risk. In this sample, the Cronbach alpha was 0.90 for the total score [33,34,35].

The Body Shape Questionnaire is a self-report measure that assesses concerns about body shape, weight, and body dissatisfaction, particularly in relation to eating disorders and body image disturbances. It includes 34 items scored at 6-point Likert scale (ranging from 1 = Never to 6 = Always), with higher scores indicating high levels of body dissatisfaction. In our sample, the Cronbach alpha was 0.98 for the total score [36,37].

The Toronto Alexithymia Scale is one of the most widely used self-report instruments for measuring alexithymia, a personality trait characterized by difficulties in identifying, describing, and processing emotions. It is particularly used in research on psychosomatic disorders, mental health, and emotional regulation. It encompasses 20 items to assess three main dimensions of alexithymia: (1) Difficulty in Identifying Feelings (DIF), that represents problems in recognizing emotions and distinguishing them from bodily sensations (an example item is “I am often confused about what emotion I am feeling”); (2) Difficulty in Describing Feelings (DDF), that represent challenges in verbally expressing emotions (an example item is: “It is difficult for me to find the right words for my feelings”); (3) Externally Oriented Thinking (EOT), that describes the tendency to focus on external, practical details rather than inner emotions (an example item is: “I prefer to analyze problems rather than just describe how I feel about them”). The questionnaire includes 20 items scored on 5-points Likert scale, with higher scores indicating high alexithymia. In our sample, the Cronbach alphas for DIF and DDF were good (0.81 and 0.83, respectively); however, EOT was excluded from the statistical analyses due to the poor reliability (α = 0.59) [38,39].

The Beck Depression Inventory is one of the most widely used self-report instruments for assessing depressive symptoms in both clinical and nonclinical populations. Each item consists of four statements ranked by severity (0 to 3 points) describing different levels of symptom severity. Respondents choose the statement that best reflects their experience in the past two weeks. Higher scores indicating high depression. In the present study, Cronbach alpha was 0.90 for the total score [40,41].

The Attachment Style Questionnaire is a self-report measure designed to assess adult attachment styles based on individual differences in interpersonal relationships. It encompasses 26 items scored at 6-points Likert scale, and evaluates attachment patterns across different relational contexts, such as romantic relationships, friendships, and social interactions. The ASQ assesses five dimensions of attachment, which correspond to insecure attachment patterns: (1) Confidence in Self and Others (Secure Attachment), that reflects comfort in relationships and trust in others (an example item is: “I find it relatively easy to get close to people”); (2) Discomfort with Closeness (Avoidant Attachment), that Measures fear of intimacy and difficulty depending on others (an example item is: “I prefer not to depend on others”); (3) Relationships as Secondary (Avoidant Attachment), that assesses tendency to prioritize independence over relationships (an example item is: “Achieving things is more important than building relationships”); (4) Preoccupation with Relationships (Anxious Attachment), that captures excessive need for reassurance and fear of abandonment (an example item is: “I worry a lot about my relationships”); (5) Need for Approval (Anxious Attachment), that measures dependence on others’ validation for self-worth (an example item is: “I worry about being alone”). In our sample, all ASQ dimensions have shown good reliability (Confidence, α = 71; Discomfort with Closeness, α = 0.81; Relationships as Secondary, α = 0.71; Preoccupation with Relationships, α = 0.68; Need for Approval, α = 0.81). In the present study, we adopted the bi-dimensional structure proposed by Fossati et al. (2003), who identified a higher-order, two-factor model representing Avoidant and Anxious attachment dimensions [42]. Within this model, the Avoidant Attachment dimension was linked to subscales such as “Discomfort with Closeness” and “Relationships as Secondary,” while the Anxious Attachment dimension included subscales like “Need for Approval” and “Preoccupation with Relationships.” Additionally, the “Confidence” subscale negatively loaded onto the Avoidant dimension, suggesting that greater confidence is associated with lower avoidance. This hierarchical structure supports the ASQ’s ability to capture both distinct attachment styles and broader attachment dimensions [42,43,44].

The Parental Bonding Instrument is a self-report questionnaire designed to assess perceived parental behaviors and bonding during childhood. It evaluates how individuals retrospectively remember their parents’ caregiving styles and control levels, which are crucial for attachment and personality development. The questionnaire includes 25 items (separately for mother and father), ranged from 0 (Very unlikely) to 3 (Very likely), and assesses parental bonding in terms of care and overprotection/control. In the current sample internal consistency ranged between 0.88 for mother protection and father protection and 0.91 for father care [22,45].

### 2.3. Statistical Analysis

All analyses were performed with SPSS for Windows 26.0. All data were checked for normality; as none of the variables reported values of skewness and kurtosis higher that 1, we used parametric analysis to verify our hypotheses. Categorical variables were described as counts and percentages, and continuous variables as mean and standard deviation. Given the threshold of 17 on the BES, statistical analyses were conducted to compare the at-risk group for moderate-to-severe BED and the non-risk group for binge eating (i.e., Non-BED risk group). A series of *t*-tests were performed for dimensional variables, while chi-square tests (*χ*^2^) and one-way Fisher exact tests were used to examine differences in contingency tables. Effect size was determined by calculating Cohen’s *d* values. Associations between variables were evaluated with Pearson’s coefficient.

To explore potential factors associated with the risk of BED, we carried out one binary logistic regression to examine the effects of sociodemographic variables (i.e., years of education; first predictor group entered into the regression model), BMI and body dissatisfaction (second predictor group), depression and alexithymia (third predictor group), attachment styles (fourth predictor group), and parental bonding (fifth predictor group) on increasing risk for BED.

All statistics were considered significant if *p* < 0.05. Occasional missing values were imputed by calculating, for each participant, the mean score of the subscale and then replaced. Adjusted odds ratios and 95% confidence intervals were calculated for the predictors of the logistic regression model.

## 3. Results

### 3.1. Sociodemographic Differences

The sociodemographic differences between individuals at risk of BED and the non-BED risk group are detailed in Table 1. Overall, no between-group differences emerged in key sociodemographic variables, i.e., sex, age, and marital status. However, participants in the non-BED risk group reported a significantly higher level of education compared to those in the at-risk of binge eating group (*p* = 0.045). Regarding clinical variables, no differences were found in regular medication use (i.e., medical comorbidity), whereas participants at high risk for binge eating reported a significantly higher BMI when compared to the non-BED risk group (*p* = 0.001).

### 3.2. Psychopathological Differences

Psychopathological between-group differences are displayed in Table 2. Most examined variables were found to be significant, except for the two PBI dimensions related to maternal and paternal overprotection/control (*p* = 0.385, *p* = 0.266, respectively). Participants at high risk of binge eating reported greater difficulty in identifying (*p* < 0.001, *d* = 0.86) and describing emotions (*p* = 0.005, *d* = 0.46), higher levels of depression (*p* < 0.001, *d* = 1.26), more anxious (*p* < 0.001, *d* = 1.04) and avoidant (*p* < 0.001, *d* = 0.62) attachment styles, and greater body dissatisfaction (*p* < 0.001, *d* = 1.91) compared to the Non-BED risk group. Additionally, individuals at high risk of binge eating scored lower on the maternal (*p* = 0.012, *d* = 0.46) and paternal (*p* = 0.017, *d* = 0.24) care dimensions of the PBI than those in non-BED risk group.

### 3.3. Correlations Between Variables

Pearson’s correlations calculated for the whole sample are shown in Table 3. Significant positive associations emerged between the total BES score and BMI (*r* = 0.34, *p* < 0.001), alexithymia dimensions (DIF: *r* = 0.33, *p* < 0.001; DDF: *r* = 0.18, *p* = 0.002), anxious (*r* = 0.41, *p* < 0.001) and avoidant (*r* = 0.27, *p* < 0.001) attachment styles, depression (*r* = 0.48, *p* < 0.001), and body image dissatisfaction (*r* = 0.61, *p* < 0.001). Conversely, significant negative correlations were found between BES scores and the parental care dimensions of the PBI, both maternal (*r* = −0.17, *p* = 0.004) and paternal (*r* = −0.15, *p* = 0.014).

### 3.4. Logistic Regression Model

A hierarchical binary logistic regression was carried out to examine the effects of sociodemographic and psychopathological variables on increasing risk for BED (Table 4). In Model 1, years of education was entered as predictor. The model was statistically significant, *χ*^2^ (1) = 4.730, *p* = 0.030, explained the 3.2% (Nagelkerke *R^2^*) of variance and correctly classified 87.6% of the cases. Years of education was found to be a significant predictor (OR = 0.859, 95%CI = 0.747–0.988, *p* = 0.033).

In Model 2, BMI, and the total score of BSQ entered as predictors. The model was significant, *χ*^2^ (2) = 75.378, *p* < 0.001, explained the 49.1% (Nagelkerke *R*^2^) of the variance and correctly classified 91.4% of the cases. In this block, only BSQ total score (OR = 1.041, 95%CI = 1.027–1.055, *p* < 0.001) emerged as statistically significant predictor, indicating that high levels of body dissatisfaction increased the risk of binge disorder of 4.1%.

In Model 3, the total BDI score and the two dimensions of alexithymia, DIF and DDF, entered as predictors. The model was not statistically significant, *χ*^2^ (8) = 14.719, *p* = 0.500, explained just an additional 0.9% (Nagelkerke *R*^2^) of the variance and correctly classified 91.4% of the cases. In this block, neither alexithymia dimensions nor the total BDI score emerged as significant predictors (DIF: *p* = 0.675; DDF: *p* = 0.676; BDI: *p* = 0.463).

In Model 4, anxious and avoidant attachment styles entered as predictors. The model was significant, *χ*^2^ (8) = 17.608, *p* = 0.024, explained an additional 2.2% (Nagelkerke *R*^2^) and correctly classified 91.4% of the cases. In this block, attachment styles did not reveal any significant association with the risk for BED (anxious attachment: *p* = 0.111; avoidant attachment: *p* = 0.083).

Finally, in Model 5, parental bonding dimensions entered as predictors. The model was not statistically significant, *χ*^2^ (8) = 15.192, *p* = 0.056, meaning that the variables added in this last step do not significantly improve the model compared to the previous blocks. This final model explained the 56.1% (Nagelkerke *R*^2^) of variance and correctly classified 92.1% of the cases. Body dissatisfaction (OR = 1.036, 95%CI = 1.019–1.054, *p* < 0.001), BMI (OR =1.155, 95%CI = 1.013–1.317, *p* = 0.031), and two dimensions of maternal bonding (i.e., maternal care: OR = 0.911, 95%CI = 0.834–0.995, *p* = 0.039, and maternal overprotection: OR = 0.895, 95%CI = 0.806–0.994, *p* = 0.039) emerged as significant predictors. Specifically, high BMI increased the risk of 15.5%, high body dissatisfaction increased the risk of 3.6%, low maternal care increased the risk of 8.9%, and low maternal protection increased the risk by 10.5% for BED.

## 4. Discussion

### 4.1. Main Findings and Study Novelty

In this critical age, we found that the 13% of participants was at risk of BED which is a serious and worrying piece of evidence. Compared to participants without BED symptoms, those at risk of BED exhibited higher BMI, lower educational attainment, and a greater prevalence of regular medication use (i.e., an indicator of medical comorbidity). They also reported higher levels of body image dissatisfaction, depressive symptoms, difficulties in identifying and describing emotions, insecure attachment styles—either avoidant or anxious—and lower maternal and paternal care during the first 16 years of life.

Overall, the group of emerging adults at risk for BED appears to be more disadvantaged and burdened, both from a sociodemographic and a psychopathological perspective.

When considering all sociodemographic and psychopathological variables simultaneously, our statistical model indicated that only higher BMI, body image dissatisfaction, and lower perception of maternal care and protection remained significant predictors of BED risk.

As we expected and it is well documented across adolescent and adult individuals [46,47], body mass index and body image dissatisfaction were related to a higher risk of BED.

In addition, the results of a large population study found that low maternal care directly contributed to body dissatisfaction [48] and childhood physical neglect enhanced body dissatisfaction in people with BED [49]. Physical neglect, defined as the caregiver’s failure to ensure basic needs and security [50], closely aligns with the low-protection dimension measured in our study.

Our original result was that lower maternal care and lower protection, describing a neglectful maternal bond, increased the risk of BED during the transitional age. We discuss this finding based on the differences among FEDs.

### 4.2. Maternal Bonding as a Unique Risk Factor and Differences Between Other Feeding and Eating Disorders

Previous studies showed that people with anorexia and bulimia nervosa reported lower levels of maternal and paternal care, while parental control was perceived as heightened [51]. Patients with FEDs recall their parents as having an “affectionless control” bond which is characterized by low warmth and empathy coupled with higher controlling attitude, restricting autonomy and independence [52].

Individuals with obesity, regardless of the presence of BED, tended to recall lower maternal and paternal care alongside higher levels of overprotection compared to those without obesity. Notably, within subgroup with obesity, the presence of BED was associated with an even lower perception of both maternal and paternal care, suggesting that reduced parental warmth may exacerbate vulnerability to binge eating beyond the effect of obesity alone [53].

Moreover, one of the studies we cited found that pre-adolescent and adolescent females (aged 12 to 16 years) with BED reported lower parental care and higher parental control [29].

Conversely, our findings indicate that, in the 18–25 age range, only the maternal bond—characterized by low care and low protection—emerged as a significant predictor of BED risk, suggesting a distinctive and age-specific vulnerability.

Anorexia and bulimia nervosa are characterized by food restriction and the debilitating control of body size aimed at building self-identity and strengthening self-efficacy, a need that may derive from an overprotective and controlling maternal bond that has limited self-esteem and autonomy [54]. Indeed, affectionless control was found to be related to self-criticism and perfectionism which in turn may lead to these eating disorders [55].

By comparison, the psychopathology of binge eating is primarily driven by emotional factors. The binge is not aimed at achieving an ideal body dimensions, at controlling the body size, but represents an impulsive and dysfunctional way of dealing with negative affect, such as depression, anxiety, and frustration, in a search for immediate gratification [56].

According to the attachment theory, the quality of the parent child bond provides a sense of self-effectiveness that, later in life, will allow an individual to regulate his emotions in stressful situations [57]. People with neglectful caregivers, who have not responded to their needs and emotional cues, will cope with distress in a framework of internalized negative experiences, thereby, developing disordered eating and excessive food intake behaviors [58]. This mechanism may be especially relevant for emerging adults who are striving to establish emotional stability and a sense of security within society [59].

In this context, the association between the perception of a neglectful mother and the presence of BED symptoms in young adults is an original result but consistent with the nature of the disorder.

The figure of a neglectful mother, rather than the controlling type of anorexia and bulimia, is a confirmation of the unique psychopathology of BED whose central core is not the need for control, but the difficulty in recognizing, managing, and regulating emotions [60], especially in this phase of life when emerging adults struggle with emotions.

### 4.3. Clinical Implications

It is concerning that 50–60% of patients with FEDs do not respond to conventional therapies for eating disorders, which predominantly target eating behavior, body weight, and body image [61,62]. This highlights the need to expand therapeutic targets to include the attachment system, given that interventions addressing attachment insecurity have been linked to improvements in BED symptoms [63].

For young adults, integrating work on early relational experiences, beliefs, and expectations about relationships may be crucial for fostering more effective emotion regulation strategies and adaptive behaviors. Individuals with avoidant attachment may experience discomfort when asked to express their emotions and feel under pressure to self-disclose them [64]. Individuals with anxious attachment focus on their own negative affect and are hypervigilant about relational losses, attempting to control anxiety by minimizing emotional distance and soliciting constant displays of support and care from others [65]. The therapist may highlight the association between emotional hyperarousal and eating disorder symptoms, encouraging the development of strategies other than binge eating to reduce negative affect [66].

Given that a suboptimal parental bond was associated with increased BED risk, preventive strategies should complement therapy by targeting parenting. Family-based and group psychoeducation could focus on fostering maternal empathy and guidance [67] while improving families’ nutritional knowledge in cases of adolescent feeding and eating disorders [68,69].

Finally, BED has been reported to reduce medication adherence in individuals with obesity [70,71]. In people with feeding and eating disorders, insecure attachment style and lower parental care have been associated with weaker therapeutic alliance and higher dropout rates from both individual and group therapy [72,73,74]. Moreover, the attachment system influences the stage of illness at which individuals decide to seek psychiatric help [75]. Therefore, integrating attachment-focused strategies into treatment is crucial to facilitate both help-seeking behavior and long-term treatment adherence.

### 4.4. Limits

We must recognize some limitations of our study. Causality in the association between mother-child bonding and risk of binge eating cannot be established, although the time span, from infancy to the transition age, is intriguing. It must be remembered that the psychometric instruments, although widely adopted in clinical study and validated, were self-reported. Nonetheless, regarding the parental bonding instrument, which provided the most interesting result, we believe that the transitional age is the ideal time to use it. Consequently, the adolescent is still too involved with the parents to evaluate the relationship and in adulthood he may have reshaped the memory of it, either because many years have passed or because he has become a parent himself. A third limitation, which afflicts most of the research on eating disorders, is the greater participation of women compared to men, preventing us from exploring a possible gender difference, especially regarding the parental bond, which is interesting.

The analysis did not stratify participants into narrower age ranges within the 18–25 group, which may conceal age-specific differences.

Another limitation of the study is the lack of information on the family’s cultural background, single-parent status, and socioeconomic status, which are factors that may influence both parental bonding and the risk of BED. Future research should include these variables to provide a more comprehensive understanding of the phenomenon.

Despite the limitations, this study provides the first strong indicator of an association between the negligent maternal bond in childhood and the development of binge eating during the transition age, with consequent therapeutic implications.

### 4.5. Future Directions

We outline that body dissatisfaction is affecting in the transitional age, when social life is intense and peer comparison influences self-esteem, life choices and contribute to self-identity [76,77].

For this reason, it would be interesting to further explore the characteristics of body dissatisfaction in young adults and to address these issues in BED therapy.

Although Bowlby stressed the importance of attachment experiences with both the father and the mother [78], we found that on a neglectful mother affected the risk of binge eating disorder. Since childhood, a low caring and controlling mother may raise disorganized and uncontrolled eating in children who fall into unregulated eating and use food as a consolation from feeling invisible, without limits or parental guidance [79]. This behavior may persist into adulthood and be exacerbated by several stressors as body dissatisfaction arising from social comparisons [80].

The effect of the maternal bond rather than that of the father has, already, been found in other studies; therefore, it is, certainly, an aspect that needs to be confirmed and clarified [81].

Finally, this study underscores the need to stimulate further research and to implement appropriate, stage-matched interventions during the transitional age; consistent with the staging model of eating disorders [82,83], such interventions are effective in halting the neuroprogression that leads to chronic, complicated, and treatment-resistant illness [84].

## 5. Conclusions

In conclusion, during the transitional period, a higher body mass index, dissatisfaction with body image, and a neglectful mother in terms of care and protection represent risk factors specific to this critical age. We argue that the treatment of binge eating disorders could benefit bringing together preventive intervention on parenting style, psychoeducation for nutrition skills, and coupling therapeutic efforts on eating disorder symptoms and attachment functioning.

## Figures and Tables

**Table 1 nutrients-17-02737-t001:** Demographic differences between Non-BED risk group and BED risk group.

Variable, M (DS)		Non-BED Risk Group	BED Risk Group	Total Sample		
		(*n* = 249)	(*n* = 38)	(*n* = 287)	*t* o *X*^2^	*p*
Age		23.25 (1.6)	23.00 (1.7)	23.21 (1.6)	*t* (47.8) = 0.90	0.392
Gender, *n* (%)					*X*^2^ (1, *n* = 287) = 1.03	0.422
	Male	65 (26.1)	7 (18.4)	72 (25.1)		
	Female	184 (73.9)	31 (81.6)	215 (74.9)		
Years of education, *n* (%)					*X*^2^ (2, *n* = 287) = 6.21	0.045
	8	2 (0.8)	0 (0.0)	2 (0.7)		
	13	86 (34.5)	21 (55.3)	107 (37.3)		
	18	161 (64.7)	17 (44.7)	178 (62.0)		
Marital status, *n* (%)					*X*^2^ (1, *n* = 287) = 1.66	0.257
	Single	236 (94.8)	34 (89.5)	270 (94.1)		
	Married	13 (5.2)	4 (10.5)	17 (5.9)		
Medication use, *n* (%)		67 (27.0)	12 (31.6)	79 (27.6)	*X*^2^ (1, *n* = 286) = 0.34	0.563
Body Mass Index		21.78 (3.0)	25.71 (6.8)	22.30 (3.9)	*t* (39.2) = −3.52	0.001

**Table 2 nutrients-17-02737-t002:** Psychopathological differences between Non-BED risk group and BED risk group.

Variable, M (DS)	Non-BED Risk Group	BED Risk Group	Total Sample	*t* o *X*^2^	*p*	Cohen’s *d*
	(*n* = 249)	(*n* = 38)	(*n* = 287)			
ASQ_Anxious Attachment	53.02 (10.6)	64.08 (10.6)	54.43 (11.24)	*t* (46.1) = −5.86	<0.001	1.04
ASQ_Avoidant Attachment	29.29 (14.1)	38.47 (15.1)	30.46 (14.5)	*t* (44.4) = −4.38	<0.001	0.62
TAS_DIF	16.52 (5.7)	19.11 (6.3)	17.15 (5.8)	*t* (43.6) = −4.77	<0.001	0.86
TAS_DDF	14.14 (5.1)	16.44 (4.2)	14.43 (5.1)	*t* (51.6) = −2.96	0.005	0.46
TAS_Total Score	48.96 (12.1)	57.11 (10.0)	50.00 (12.1)	*t* (51.3) = −4.42	<0.001	0.69
BDI_Total Score	9.00 (7.7)	19.29 (10.8)	10.39 (8.9)	*t* (43.0) = −5.66	<0.001	1.26
BSQ_Total Score	77.77 (33.3)	142.74 (38.3)	85.95 (40.2)	*t* (41.7) = −9.52	<0.001	1.91
PBI_Mother_Care	38.13 (7.3)	34.22 (8.5)	37.64 (7.5)	*t* (42.9) = 2.62	0.012	0.53
PBI_Mother_Protection	25.37 (7.5)	26.64 (8.2)	25.55 (7.6)	*t* (44.4) = −0.88	0.385	0.17
PBI_Father_Care	34.93 (8.4)	31.03 (8.6)	34.44 (8.5)	*t* (42.4) = 2.53	0.017	0.46
PBI_Father_Protection	22.89 (6.8)	24.53 (8.1)	23.07 (7.0)	*t* (39.2) = −1.28	0.266	0.24

Note. ASQ = Attachment Style Questionnaire; TAS_DIF = Toronto Alexithymia Scale—Difficulty in Identifying Feelings; TAS_DDF = Toronto Alexithymia Scale—Difficulty in Describing Feelings; BDI = Beck Depression Inventory; BSQ = Body Shape Questionnaire; PBI = Parental Bonding Instrument.

**Table 3 nutrients-17-02737-t003:** Correlations between variables.

		1	2	3	4	5	6	7	8	9	10	11	12
1	BMI	1											
2	ASQ_Anxious Attachment	0.09											
3	ASQ_Avoidant Attachment	0.15 *	0.54 ***										
4	TAS_DIF	0.10	0.47 ***	0.45 ***									
5	TAS_DDF	0.11	0.27 ***	0.53 ***	0.54 ***								
6	TAS_Total Score	0.13 *	0.37 ***	0.55 ***	0.80 ***	0.84 ***							
7	BDI_Total Score	0.20 **	0.57 ***	0.54 ***	0.56 ***	0.34 ***	0.46 ***						
8	BSQ_Total Score	0.37 ***	0.45 ***	0.35 ***	0.38 ***	0.19 **	0.28 ***	0.56 ***					
9	PBI_Mother_Cure	−0.05	−0.26 ***	−0.18 **	−0.20 **	−0.12 *	−0.15 *	−0.21 ***	−0.13 *				
10	PBI_Mother_Protection	0.07	0.20 **	0.16 **	0.22 ***	0.09	0.16 **	0.20 **	0.11	−0.56 ***			
11	PBI_Father_Cure	−0.04	−0.36 ***	−0.26 ***	−0.20 **	−0.22 ***	−0.21 ***	−0.25 ***	−0.13 *	0.46 ***	−0.36 ***		
12	PBI_Father_Protection	−0.02	0.18 **	0.14 *	0.16 **	0.14 *	0.19 **	0.17 **	0.09	−0.43 ***	0.57 ***	−0.49 ***	
13	BES_Total Score	0.34 ***	0.41 ***	0.27 **	0.33 ***	0.18 **	0.27 ***	0.48 ***	0.61 ***	−0.17 **	0.09	−0.15 *	0.08

Note. * *p* < 0.05; ** *p* < 0.01; *** *p* < 0.001. ASQ = Attachment Style Questionnaire; TAS_DIF = Toronto Alexithymia Scale—Difficulty in Identifying Feelings; TAS_DDF = Toronto Alexithymia Scale—Difficulty in Describing Feelings; BDI = Beck Depression Inventory; BSQ = Body Shape Questionnaire; PBI = Parental Bonding Instrument; BES = Binge Eating Scale.

**Table 4 nutrients-17-02737-t004:** Binary logistic regression analysis predicting risk of BED vs. non-BED risk based on demographic characteristics, body dissatisfaction, alexithymia, depression, attachment styles, and parental bonding.

	Model 1			Model 2			Model 3			Model 4			Model 5		
	OR	95%CI	Wald	OR	95%CI	Wald	OR	95%CI	Wald	OR	95%CI	Wald	OR	95%CI	Wald
Years of education	0.859	0.747–0.988	4.565 *	0.909	0.761–1.085	1.114	0.942	0.775–1.144	0.366	0.963	0.792–1.172	0.138	0.925	0.750–1.142	0.523
BSQ_Total Score				1.041	1.027–1.055	35.557 ***	1.035	1.019–1.052	0.008 ***	1.035	1.018–1.052	17.437 ***	1.036	1.019–1.054	17.923 ***
BMI				1.115	0.994–1.249	3.461	1.127	1.000–1.271	0.061 *	1.134	1.008–1.277	4.343 *	1.155	1.013–1.317	4.642 *
BDI_Total Score							1.023	0.962–1.088	0.031	1.034	0.966–1.107	0.913	1.039	0.970–1.113	1.172
TAS_DIF							1.023	0.921–1.135	0.053	0.991	0.888–1.107	0.023	1.022	0.913–1.144	0.145
TAS_DDF							1.023	0.918–1.141	0.056	1.082	0.955–1.225	1.537	1.059	0.929–1.208	0.738
ASQ_Anxious Attachment										1.052	0.988–1.119	2.536	1.034	0.965–1.107	0.895
ASQ_Avoidant Attachment										0.958	0.912–1.006	2.996	0.962	0.915–1.012	2.232
PBI_Mother_Cure													0.911	0.834–0.995	4.281 *
PBI_Mother_Protection													0.895	0.806–0.994	4.269 *
PBI_Father_Cure													0.985	0.911–1.064	0.156
PBI_Father_Father													1.000	0.906–1.103	0.000

Note. * *p* < 0.05; *** *p* < 0.001. ASQ = Attachment Style Questionnaire; TAS_DIF = Toronto Alexithymia Scale—Difficulty in Identifying Feelings; TAS_DDF = Toronto Alexithymia Scale—Difficulty in Describing Feelings; BDI = Beck Depression Inventory; BSQ = Body Shape Questionnaire; PBI = Parental Bonding Instrument.

## Data Availability

The dataset is available from the corresponding author upon reasonable request due to ethical reasons.

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
