# Peer review of "Low Maternal Care and Protection and Body Image Dissatisfaction as Psychopathological Predictors of Binge Eating Disorder in Transitional-Age Youth"

_nutrients, 2025, doi:10.3390/nu17172737_

Round 1
Reviewer 1 Report
Comments and Suggestions for Authors
an interesting and informative study
here are some comments, questions, and recommendations
abstract - might go over the maximum limit in the number of words
would suggest to remove the statistical information (eg OR = 1.036, 95%CI = 1.019-1.054, p < .001) author might say, that
the final model explained the ..... with body dissatisfaction, BMI.... as significant predictors...
line 51 - ...onset of approximately 75% of mental disorders occurs before the age...
would suggest to include more details - such as .... according to a meta-analysis of ... studies from (inclusive years).... mental disorders..
adulthood is a major life transition probably representing the most unstable period of their lifespan - this should be cited
parental bonding - lines 78 to 81 should be cited
although few studies are noted - as the author/s presented - would probably expand later in the literature review
what parental bonding would likely affect to the adolescent stage
and how these in turn might affect binge eating behavior
aim of the study - might provide some hypotheses? Research questions?
no literature review?
should provide some background of the concepts / factors used or hypothesized to be related to binge eating
inclusion and exclusion criteria
research locale?
why these measures?
why these demographics?
data collection procedure?
there should be some linkage between the measures and a wider theoretical concept or framework
study is very descriptive, since there are no hypotheses provided
direction of discussions is somewhat hard to understand
would help if there are subheadings
what does the group differences meant
what now? practical implications
Author Response
We thank the reviewers for their thoughtful and constructive comments. Below, we address each point in detail. For clarity, the reviewers’ comments are reported in italics, followed by our responses. All corresponding changes in the manuscript are highlighted in the revised version.
Reviewer 1
an interesting and informative study
here are some comments, questions, and recommendations
- abstract - might go over the maximum limit in the number of words
would suggest to remove the statistical information (eg OR = 1.036, 95%CI = 1.019-1.054, p < .001) author might say, that
the final model explained the ..... with body dissatisfaction, BMI.... as significant predictors...
Thank you for pointing this out. Therefore, we have revised the abstract. Please see lines 13-34
- line 51 - ...onset of approximately 75% of mental disorders occurs before the age...
would suggest to include more details - such as .... according to a meta-analysis of ... studies from (inclusive years).... mental disorders..
We agree with this comment. Therefore we included the details you suggested from lines 46-48.
- adulthood is a major life transition probably representing the most unstable period of their lifespan - this should be cited
We thank the reviewer for this comment. Accordingly, we have added Reference No. 7 to the manuscript.
- parental bonding - lines 78 to 81 should be cited
We agree with this comment. Accordingly, we have added References No. 19, 20, and 21 to the manuscript.
- although few studies are noted - as the author/s presented - would probably expand later in the literature review
what parental bonding would likely affect to the adolescent stage
and how these in turn might affect binge eating behavior
We are grateful to the reviewer for allowing us to improve this point, which required further clarification. In particular, we added a review on the association between parental bonding and eating disorder (ref. 25) improving the section related to how parental bonding may affect adolescent binge eating (lines 81-90) and describing that low parental care and high overprotection care were associated with eating disorders
- aim of the study - might provide some hypotheses? Research questions?
We thank the reviewer, and we provided a more detailed hypothesis as shown in line 91-98
- no literature review?
should provide some background of the concepts / factors used or hypothesized to be related to binge eating
We again thank the reviewer and have followed the suggestion, as described in our response to Q5.
- inclusion and exclusion criteria
research locale?
We thank the reviewer. Following the suggestion, we clarify inclusion and exclusion criteria (line 118-125). The research was local as we wrote in the revised manuscript (line 119).
- why these measures?
why these demographics?
We are grateful to the reviewer for allowing us to improve this point, from lines 55 to 58.
We also elucidated demographics at line 127-131
- data collection procedure?
Thank you for pointing this out, we followed your suggestion at line 108-113
- there should be some linkage between the measures and a wider theoretical concept or framework
Thank you for pointing this out. We revised the Introduction, increasing the background of our investigation, removing redundant parts, and improving references.
- study is very descriptive, since there are no hypotheses provided
direction of discussions is somewhat hard to understand
would help if there are subheadings
We appreciate the reviewer’s suggestion. We clarified the hypotheses. As highlighted in the red-marked sections, we have reformulated the Discussion to improve its logical flow, removed redundant parts, added subheadings, and expanded the references.
- what does the group differences meant
We thank the reviewer. As group differences refer to the statistical comparisons between the BED risk group (BES score ≥ 17) and the non-BED risk group (BES score < 17), we revised the tables and results to improve clarity. We addressed these differences in the Discussion section (lines 296-305).
- what now? practical implications
We thank the reviewer. In the “clinical implication” section of the discussion we described practical implications (lines 370-399)
Reviewer 2 Report
Comments and Suggestions for Authors
This is a topic of great interest, and the approach to addressing it is what arouses curiosity.
Introduction
There are some elements that must be addressed, even if only superficially, such as the cultural roots of the family, whether they are single-parent families, and their socioeconomic status. All these elements can influence the final results.
It would be useful to indicate, at a general level, what types of tools are used to measure these variables so that the reader can obtain an overall idea of the situation.
The hypotheses of this study should be reflected.
Methods
No es posible que los datos se recojan entre septiembre y diciembre de 2025, imagino que es 2024, corrijan estopor favor.
The age range is a little high, as they are of legal age, and perhaps some are far from their family environment, so the choice of this age range would need to be well justified.
Discussion
The survey failed to highlight the differences between the sexes, which is relevant.
Within the limitations, it should be added that the analysis, in addition to the distinction between sexes, should also distinguish between age ranges. There are significant differences between the ages of 18 and 25 years, which have not been discussed.
The study sample was small, and it may be appropriate to describe the study as exploratory. Therefore, the tone in which the results are discussed should be more cautious.
Author Response
We thank the reviewers for their thoughtful and constructive comments. Below, we address each point in detail. For clarity, the reviewers’ comments are reported in italics, followed by our responses. All corresponding changes in the manuscript are highlighted in the revised version.
Reviewer 2
This is a topic of great interest, and the approach to addressing it is what arouses curiosity.
Introduction
- There are some elements that must be addressed, even if only superficially, such as the cultural roots of the family, whether they are single-parent families, and their socioeconomic status. All these elements can influence the final results.
We appreciate the reviewer’s observation. Unfortunately, information regarding the cultural background of the family, single-parent status, and socioeconomic status was not collected in the present study, and we therefore acknowledge this as a limitation (lines 415-418). These factors are indeed relevant, as they may influence both parental bonding patterns and the risk of BED. They should be considered in future research to provide a more comprehensive understanding of the phenomenon.
- It would be useful to indicate, at a general level, what types of tools are used to measure these variables so that the reader can obtain an overall idea of the situation.
We thank the reviewer- Following this suggestion we detailed the Methods (lines 127-131)
- The hypotheses of this study should be reflected.
We appreciate your suggestion, we clarified background and hypotheses (lines 91-98)
Methods
- No es posible que los datos se recojan entre septiembre y diciembre de 2025, imagino que es 2024, corrijan estopor favor.
We thank the reviewer for pointing it out. We corrected the sentence (lines 108-113)
- The age range is a little high, as they are of legal age, and perhaps some are far from their family environment, so the choice of this age range would need to be well justified.
We thank the reviewer for this comment. The age range of 18–25 years was chosen to specifically target the transitional age, a developmental stage between adolescence and adulthood characterized by ongoing psychological, social, and relational changes. Even though individuals are legally adults, many remain emotionally and, in part, practically connected to their family of origin. This age range has been widely used in the literature to study emerging adulthood and its associated vulnerabilities, including those related to eating disorders (Arnett, J.J. et al, ref. no.8). To emphasize this point, we have added reference no. 7 to support the rationale for selecting this age range and have made it explicit in the Methods section (lines 104–106).
Discussion
- The survey failed to highlight the differences between the sexes, which is relevant.
We agree with the reviewer’s comment. We included this argument in the Limitations section (lines 417-420)
- Within the limitations, it should be added that the analysis, in addition to the distinction between sexes, should also distinguish between age ranges. There are significant differences between the ages of 18 and 25 years, which have not been discussed.
We thank the reviewer for this observation. We acknowledge that there may be relevant differences within the 18–25 age range, and that our analysis did not stratify participants by narrower age groups. We added this consideration to the Limitations section, noting that future studies should examine potential age-related differences within the transitional period. (lines 415-416)
- The study sample was small, and it may be appropriate to describe the study as exploratory. Therefore, the tone in which the results are discussed should be more cautious.
We appreciate the reviewer’s suggestion and have tempered the Discussion and expanded the Limitations section accordingly.
Round 2
Reviewer 1 Report
Comments and Suggestions for Authors
after going over the point by point revisions made by the authors, the paper is now better and acceptable